# The MsrAB reducing pathway of *Streptococcus gordonii* is needed for oxidative stress tolerance, biofilm formation, and oral colonization in mice

Naif Jalal[1,2¤], Song F. Lee[1,2,3]*

**1** Department of Microbiology and Immunology, Dalhousie University, Halifax, Nova Scotia, Canada, **2** Canadian Center for Vaccinology, Dalhousie University, Nova Scotia Health Authority, Izaak Walton Killam Health Centre, Halifax, Nova Scotia, Canada, **3** Department of Applied Oral Sciences, Faculty of Dentistry, Dalhousie University, Halifax, Nova Scotia, Canada

¤ Current address: Department of Medical Microbiology, Faculty of Medicine, Umm Al-Qura University, Makkah, Saudi Arabia
* Song.Lee@Dal.Ca

**Data Availability Statement:** All relevant data are within the paper and its Supporting Information files.

## Abstract

The ability of *Streptococcus gordonii* to cope with oxidative stress is important for survival and persistence in dental plaque. In this study, we used mutational, phenotypic, and biochemical approaches to characterize the role of a methionine sulfoxide reductase (MsrAB) and proteins encoded by genes in the *msrAB* operon and an adjacent operon in oxidative stress tolerance in *S. gordonii*. The results showed that MsrAB and four other proteins encoded in the operons are needed for protection from $H_2O_2$ and methionine sulfoxide. These five proteins formed a reducing pathway that was needed for oxidative stress tolerance, biofilm formation, and oral colonization in mice. In the pathway, MsrAB was the enzyme that repaired oxidatively damaged proteins, and the two thioredoxin-like lipoproteins (SdbB and Sgo_1177) and two CcdA proteins were proteins that maintained the catalytic cycle of MsrAB. Consistent with the role in oxidative stress tolerance, the production of MsrAB, SdbB, and Sgo_11777 was induced in aerobic growth and planktonic cells.

## Introduction

*Streptococcus gordonii* is a pioneer colonizer of the human tooth surface and plays an important role in dental plaque (dental biofilm) formation [1]. The bacterium binds to salivary components coated on the tooth surface forming the first layer of cells on the tooth. It then coaggregates with other oral bacteria promoting the development of a multi-layer dental biofilm. $H_2O_2$, a byproduct of aerobic metabolism, is produced by many streptococci, including *S. gordonii* [2, 3]. The level of $H_2O_2$ is reported to reach mM concentration in oral bacterial biofilms [4]. As a powerful oxidant, $H_2O_2$ can damage proteins [5]. In particular, methionine is oxidized to methionine sulfoxide, which can be further oxidized to methionine sulfone, an irreversible modification. The oxidation, if not repaired, can lead to loss of function or

**Funding:** Funding for this study was provided by the Natural Sciences and Engineering Research Council of Canada (NSERC) grant #183712 to SFL. NJ is a recipient of a postgraduate scholarship from the Saudi Cultural Bureau and Umm Al-Qura University. The funders had no role in study design, data collection and analysis, decision to publish, or preparation of the manuscript.

**Competing interests:** The authors have declared that no competing interests exist.

degradation of proteins [5]. Thus, to be successful as an inhabitant of the dental plaque, *S. gordonii* needs to counter the oxidative effects of $H_2O_2$. *S. gordonii* produces a superoxide dismutase but lacks a catalase to decompose $H_2O_2$ [6, 7]

Methionine sulfoxide reductases are enzymes that reduce methionine sulfoxide to methionine [8, 9]. In many bacteria, the reduction is an important mechanism to repair oxidatively damaged proteins [8]. A methionine sulfoxide reductase A (MsrA or Sgo_0278) was previously described in *S. gordonii* [10, 11]. MsrA is 36 kDa protein. Western blotting and immunofluorescent analysis showed that only trace amount of MsrA was detected in the cell wall fraction and cell surface, respectively, and the majority of the protein was located in the cytoplasm, indicating that MsrA is a cytoplasmic protein [10]. *msrA* mutants were sensitive to $H_2O_2$ and defective in adhesion. *msrA* mutants were defective in biofilm formation but only in the presence of exogenous $H_2O_2$ [10].

The genome of *S. gordonii* encodes another methionine sulfoxide reductase named MsrAB (Sgo_1176), which is located on a different locus from *msrA* [12]. MsrAB is a single polypeptide with a predicted size of 42.5 kDa and contains a signal sequence suggesting that its cell location is extracytoplasmic. It is annotated as MsrAB presumably it contains both the A and B catalytic domains. The biological function of MsrAB has not been investigated. In this study, we showed that MsrAB plays an important role in protecting *S. gordonii* from oxidative stress and confers fitness in an animal colonization model. We showed that *msrAB* mutants were defective in biofilm formation in the absence of exogenous $H_2O_2$. We presented results to suggest a pathway for the regeneration of MsrAB. This pathway involves four other proteins located in two adjacent operons. This pathway represents a reducing pathway in *S. gordonii*, which is the first such pathway reported for this organism.

## Results

### MsrAB, SdbB, Sgo_1177, CcdA1 and CcdA2 are involved in oxidative stress tolerance

We previously reported that *msrAB* is the last gene in a five gene-operon coding for a putative two-component regulatory system (Sgo_1180 and Sgo_1181), CcdA1 (cytochrome *c* biogenesis protein A), a thioredoxin-like lipoprotein (Sgo-1177), and MsrAB (S1 Fig) [13]. Immediately downstream from *msrAB* is a four-gene operon for another putative two-component regulatory system (Sgo_1174 and Sgo_1175), a second cytochrome *c* biogenesis protein A (CcdA2), and a thioredoxin-like lipoprotein SdbB. We previously showed that CcdA2 and SdbB are redox partners of the thiol-disulfide oxidoreductase SdbA [13].

To help to assess if *msrAB* and neighboring genes played a role in oxidative stress tolerance, single- and double-gene knockout mutants were constructed (Table 1). In addition, a *msrA* (*sgo_0278*) mutant was also constructed for comparison as this mutant was reported to be sensitive to $H_2O_2$ [10]. Initially, the mutants were tested for sensitivity to $H_2O_2$. The results showed that the *msrAB* mutant was more sensitive to $H_2O_2$ than the parent strain (Fig 1A). The *msrA* mutant was also sensitive to $H_2O_2$, consistent with the previous report [10]. Interestingly, other single-gene mutants, namely *sdbB*, *sgo_1177*, *ccdA1*, and *ccdA2* mutants, were also sensitive to $H_2O_2$. The double-gene mutants were also more sensitive to $H_2O_2$ compared to the parent strain with the cross-operon double-gene mutants, *sdbBsgo_1177* and *ccdA1ccdA2*, being the most sensitive showing a ~1000-fold reduction in survival. Complementation of *sgo_1177ccdA1* and *sdbBccdA2* restored the level of sensitivity to that of the parent strain.

Next, we tested the mutants for sensitivity to methionine sulfoxide. All *S. gordonii* strains grew to a similar optical density in media without methionine sulfoxide (Fig 1B). In the presence of methionine sulfoxide, the growth of all gene-knockout mutants, including *msrAB* and

**Table 1. Bacterial strains and plasmids used in this study.**

| Strains/plasmids | Relevant characteristics | Source |
|---|---|---|
| ***S. gordonii*** **Challis DL-1** | | |
| SecCR1 | *hppG::tet*, secretes anti-CR1 scFv, Tet[R], Spec[R] | [29] |
| *ccdA1* | CcdA1-negative mutant in SecCR1, *ccdA1::aphA3* Tet[R], Spec[R], Kan[R] | [13] |
| *ccdA2* | CcdA2-negative mutant in SecCR1, *ccdA2::ermAM* Tet[R], Spec[R], Erm[R] | [13] |
| *sdbB* | SdbB-negative mutant in SecCR1, *sgo_1171::ermAM*, Tet[R], Spec[R], Erm[R] | [13] |
| *sgo_1177* | SGO_1177-negative mutant in SecCR1, *sgo_1177::ermAM*, Tet[R], Spec[R], Erm[R] | [13] |
| *msrAB* | MsrAB-negative mutant in SecCR1, *sgo_1176::ermAM*, Tet[R], Spec[R], Erm[R] | This study |
| *msrA* | MsrA-negative mutant in SecCR1, *sgo0278::aphA3*, Tet[R], Spec[R], Kan[R] | This study |
| *sgo_1177ccdA1* | SGO_1177 and CCdA1-negative mutant in SecCR, *sgo_1177::ermAM*, *ccdA1::aphA3*, Tet[R], Spec[R], Kan[R],Erm[R] | [13] |
| *sgo_1177ccdA1 complement* | SecCR1, *ccdA1sgo_1177* complemented on the chromosome, Tet[R], Spec[R], Cm[R] | [13] |
| *sdbBccdA2* | SdbB and CcdA2-negative mutant in SecCR1, *sgo_1171::aphA3*, *ccdA2::ermAM*, Tet[R], Spec[R], Kan[R],Erm[R] | [13] |
| *sdbBccdA2 complement* | SecCR1, *ccdA2sgo_1171* complemented on the chromosome, Tet[R], Spec[R], Cm[R] | [13] |
| ***E. coli*** | | |
| M15 | Protein expression host, pRE4, Kan[R] | Qiagen |
| SdbB | M15 carrying pQE30-*sdbB*, Kan[R], Amp[R] | [13] |
| Sgo_1177 | M15 carrying pQE30-*sgo_1177*, Kan[R], Amp[R] | [13] |
| XL-1 Blue | Cloning host, Tet[R] | Stratagene |
| CcdA1 | XL-1 Blue carrying pQE30-*ccdA1*, Tet[R], Amp[R] | This study |
| CcdA2 | XL-1 Blue carrying pQE30-*ccdA2*, Tet[R], Amp[R] | [13] |
| MsrAB | XL-1 Blue carrying pQE30-*msrAB*, Tet[R], Amp[R] | This study |
| MsrAB-A | XL-1 Blue carrying pQE30-*msrAB-A*, Tet[R], Amp[R] | This study |
| MsrAB-B | XL-1 Blue carrying pQE30-*msrAB-B*, Tet[R], Amp[R] | This study |
| **Plasmids** | | |
| pQE30SdbB | *sdbB* cloned into BamHI and HindIII sites on pQE30, His$_6$-tag, Amp[R] | [13] |
| pQE30Sgo_1177 | *sgo_1177* cloned into BamHI and HindIII sites on pQE30, His$_6$-tag, Amp[R] | [13] |
| pQE30CcdA1 | *ccdA1* cloned into BamHI and EcoRV sites on pQEDegP, N- and C-terminal His$_6$, C-terminal HA-tag | This study |
| pQE30CcdA2 | *ccdA2* cloned into BamHI and EcoRV sites on pQEDegP, N- and C-terminal His$_6$, C-terminal HA-tag | [13] |
| pQE30MsrAB | *msrAB* cloned into BamHI and HindIII sites on pQE30, His$_6$-tag, Amp[R] | This study |
| pQE30MsrAB-A | The A domain of *msrAB* cloned into BamHI and HindIII sites on pQE30, His$_6$-tag, Amp[R] | This study |
| pQE30MsrAB-B | The B domain of *msrAB* cloned into BamHI and HindIII sites on pQE30, His$_6$-tag, Amp[R] | This study |

*msrA* mutants, were significantly more impaired than the parent strain (Fig 1C). Complementation of *sgo_1177ccdA1* and *sdbBccdA2* restored the sensitivity in the double-gene mutants to that of the parent strain.

## The catalytic cycle of MsrAB

The above results indicate that MsrAB, SdbB, Sgo_1177, CcdA1, and CcdA2 played a role in protecting *S. gordonii* from the toxic effects of $H_2O_2$ and methionine sulfoxide. To provide biochemical evidence of the above statement, we performed the methionine sulfoxide reductase assay. Recombinant proteins were produced and purified from *E. coli* (S2 Fig). In the assay, MsrAB activity was determined using methionine sulfoxide as the substrate and NADPH as the source of electrons. Thioredoxin reductase (TrxB) transfers electrons from NADPH to a test protein (SdbB, Sgo_1177, CcdA1, or CcdA2), which in turn passes the electrons to MsrAB, which then reduces methionine sulfoxide. The reduction of methionine sulfoxide by MsrAB was monitored at $A_{340}$ as the oxidation of NADPH. The results showed that MsrAB reduced methionine sulfoxide in the presence of SdbB or Sgo_1177 (Fig 2A). No MsrAB activity was

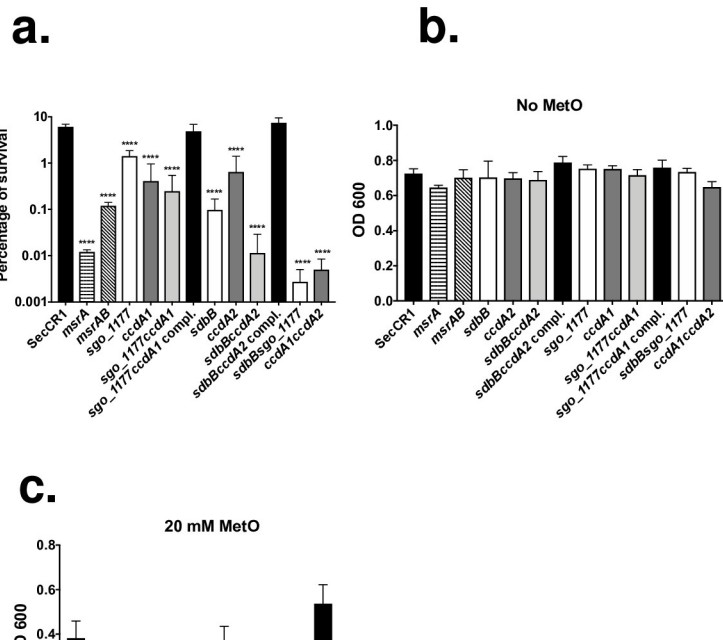

**Fig 1. Sensitivity to H$_2$O$_2$ and methionine sulfoxide by *S. gordonii*.** (a) Sensitivity of *S. gordonii* to H$_2$O$_2$. *S. gordonii* were challenged with 10 mM H$_2$O$_2$ for 30 minutes, and the percentage of survival was determined by CFU counts. Data are mean ± SD of three independent experiments with triplicate in each experiment. (b and c) Sensitivity of *S. gordonii* to methionine sulfoxide (MetO). *S. gordonii* were grown for 24 hours in media alone (b) or with 20 mM methionine sulfoxide (c). Results are means ± SD of two independent experiments with duplicate in each experiment. (***P < 0.001; ****P < 0.0001).

detected when SdbB or Sgo_1177 (Fig 2A) or MsrAB (S3 Fig) was omitted in the reaction. The specific activity of MsrAB with SdbB or Sgo_1177 was 23.2 ± 1.6 or 24.7 ± 1.6 nmol NADPH/ mg MsrAB/min (*P* = 0.902), respectively. When CcdA1 or CcdA2 was used in place of SdbB or Sgo_1177 in the reaction, no MsrAB activity was detected (Fig 2B and 2C).

Since MsrAB contains two catalytic domains, A and B, it is possible that SdbB and Sgo_1177 have a differential preference for these domains. To test this, recombinant "A" and "B" domain of MsrAB were produced and purified (S2 Fig) and tested in the methionine sulfoxide reductase assay. The results showed that the A and B domains of MsrAB were active in the presence of SdbB or Sgo_1177 (Fig 2D and 2E). These domains lacked activity when SdbB or Sgo_1177 was omitted in the reaction. The specific activity of the "A' domain with SdbB or Sgo_1177 was 24.7 ± 2.5 or 25.8 ± 2.3 nmol NADPH/mg A/min (*P* = 0.978), respectively. The specific activity of the "B' domain with SdbB or Sgo_1177 was 24.55 ± 1.8 or 23.62 ± 1.7 nmol NADPH/mg B/min (*P* = 0.944), respectively.

The above results suggest that SdbB and Sgo_1177, not CcdA1 and CcdA2, are the immediate partner of MsrAB in the reduction of methionine sulfoxide. To provide further support of this notion, we conducted disulfide exchange reactions between these proteins. We reasoned that in order for the catalytic cycle to continue, oxidized MsrAB generated from methionine sulfoxide reduction needs to be reduced by a partner. We hypothesize that SdbB and Sgo_1177

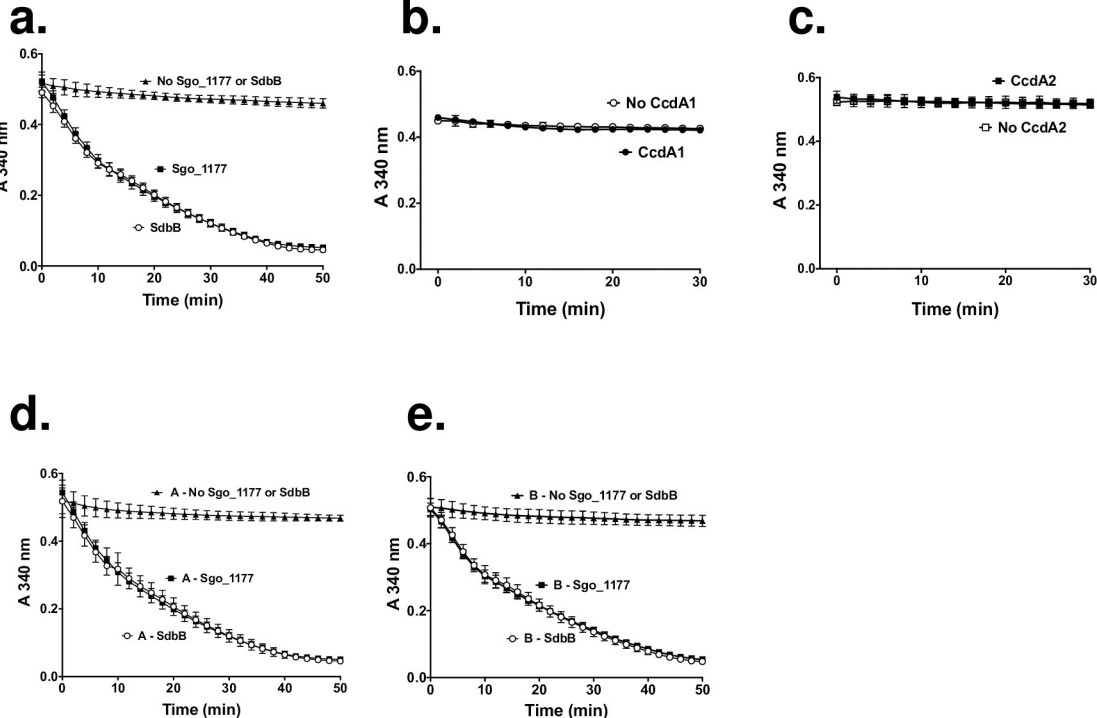

**Fig 2. Methionine sulfoxide reductase activity of MsrAB.** (a) MsrAB activity in the presence of SdbB or Sgo_1177. (b) MsrAB activity in the presence of CcdA1. (c) MsrAB activity in the presence of CcdA2. (d) Activity of A domain of MsrAB in the presence of SdbB or Sgo_1177. (e) Activity of B domain of MsrAB in the presence of SdbB or Sgo_1177. In all panels, reactions without test proteins (SdbB, Sgo_1177, CcdA1, or CcdA2) were included as controls. Results are means ± SD of three independent experiments with duplicate in each experiment.

are the immediate redox partners of MsrAB. To facilitate detection, we generated a polyclonal antibody to MsrAB, and the antibody recognized both the reduced and oxidized forms of MsrAB and showed no cross-reaction to SdbB or Sgo_1177 (Fig 3A). The results showed that oxidized MsrAB was rapidly reduced by SdbB (Fig 3B) or Sgo_1177 (Fig 3C). Consistent with results of the methionine sulfoxide reductase assay, oxidized MsrAB was not reduced by CcdA1 or CcdA2 (Fig 3D).

Although CcdA1 and CcdA2 were not able to reduce MsrAB, they could still play a role in the catalytic cycle by reducing SdbB and Sgo_1177, which in turn reduce MsrAB. To test this, disulfide exchange reactions between these proteins were performed. The results showed that CcdA1 could reduce SdbB and Sgo_1177 (Fig 3E and 3F). CcdA2 was also capable of doing the same.

The above results suggest a pathway for the catalytic cycle of MsrAB: SdbB and Sgo_1177 are the immediate partners of MsrAB regenerating reduced MsrAB; CcdA1 and CcdA2 are the partners of SdbB and Sgo_1177 regenerating reduced SdbB and Sgo_1177. The pathway allows MsrAB to continue the reduction of methionine sulfoxide.

## MsrAB is an extracytoplasmic enzyme and required for biofilm formation and oral colonization in mice

Analysis of the MsrAB sequence using SignalP 4.0 [14] revealed that the protein carries a signal sequence. This is similar to MsrAB2 previously described in *Streptococcus pneumoniae*, which also carries a well-defined signal sequence (Fig 4A) [15]. In contrast, the previously described

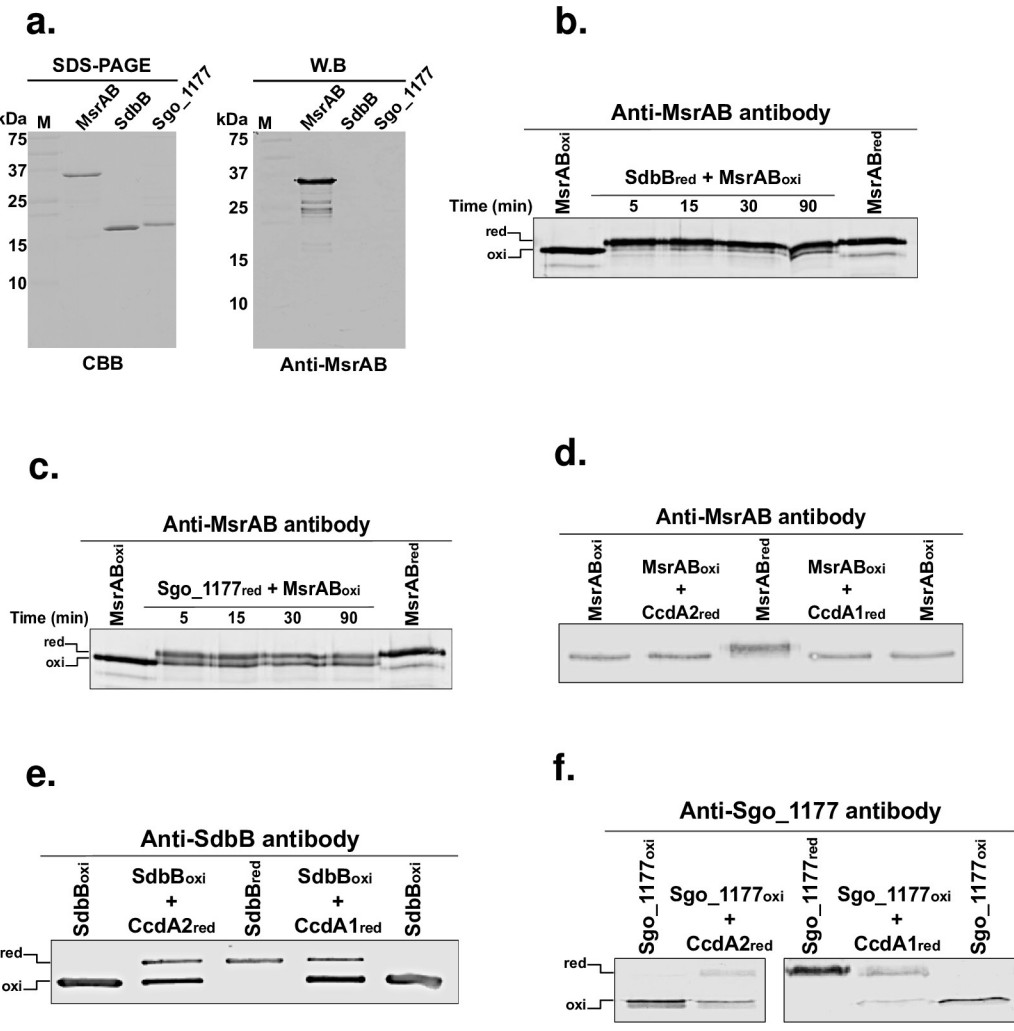

**Fig 3. Reduction of MsrAB by SdbB and Sgo_1177.** (a) SDS-PAGE and immunoblotting (WB) of purified recombinant MsrAB, SdbB, and Sgo_1177. M: prestained protein markers. (b) Reduction of oxidized MsrAB by reduced SdbB. (c) Reduction of oxidized MsrAB by reduced Sgo_1177. (d) Lack of reduction of oxidized MsrAB by reduced CcdA1 or CcdA2. (e) Reduction of oxidized SdbB by reduced CcdA1 or CcdA2. (f) Reduction of oxidized Sgo_1177 by reduced CcdA1 or CcdA2. In panels d, e, and f, disulfide exchange reactions were carried out for 30 minutes. Antibodies used were mouse antisera: anti-MsrAB (1/500), anti-SdbB (1/1000) and anti-Sgo_1177 (1/1000).

MsrA of *S. gordonii* lacks a signal sequence. To provide evidence that MsrAB is an extracytoplasmic enzyme, *S. gordonii* was separated into cell wall and cell membrane-cytoplasm fractions and probed with the anti-MsrAB antibody. As shown in Fig 4B, a *ca*. 38 kDa band, the expected size of the mature MsrAB, was detected in the cell wall fraction but not in the cell membrane-cytoplasm fraction, supporting the notion that MsrAB is an extracytoplasmic protein.

Next, we examined the ability of the *msrAB* mutant to form biofilm. We reasoned that as an extracytoplasmic enzyme, MsrAB might play a role in repairing oxidatively damaged surface proteins important to biofilm formation. We included the *msrA* mutant in the assay for comparison because this mutant was defective in biofilm formation in the presence of exogenous $H_2O_2$ [10]. As shown in Fig 5A, the *msrAB* mutant formed less biofilm compared to the parent and *msrA* mutant strains. The *msrA* mutant showed no defect in biofilm formation under our

a.

```
SgMsrA    ------------------------------------------------------------    0
SgMsrAB   MESKWKMIMFILAFLLCFGLFFLIRGSMSIDSSHANAEQIKKASMSKTEVVKKKKEDVKE   60
SpMsrAB2  MNDKLKIFLLLGVFFLAITGFYVLL---IRNAGQTDASQIEKAAVSQGGKAVKKTEISKD   57

SgMsrA    --MAEIYLAGGCFWGLEEYFSRIEGVKKTTVGYANGQVESTNYQLIHQTDHAETVHLIYD   58
SgMsrAB   ADKRVIYLAGGCFWGVEEYFSRVPGVIDAESGYANGKGDTTKYELVSQTGHAETVKITYN  120
SpMsrAB2  ADLHEIYLAGGCFWGVEEYFSRVPGVTDAVSGYANGRGETTKYELINQTGHAETVHVTYD  117
            **********:******: **  .:  *****: ::*:*:*: **.******:: *:

SgMsrA    EKRVSLREILLYYFRVIDPLSVNKQGNDVGRQYRTGVYYTNQADKAVIEQVFAEQEKQLG  118
SgMsrAB   VKKISLKEILLHYFRIIDPTSKNKQGNDQGTQYRTGVYYKDEADLNTINQVFDEVAKKYD  180
SpMsrAB2  AKQISLKEILLHYFRIINPTSKNKQGNDVGTQYRTGVYYTDDKDLEVINQVFDEVAKKYD  177
           *::**:****:***:*:* * ****** * ********.:: *  .*:*** *  *:.

SgMsrA    QKIAVELEPLRHYVLAEDYHQDYLKKNPGGYCHINVNDAYQPLVDPGQYEKPTDAELKEQ  178
SgMsrAB   KPLAVEKEPLKNYVKAENYHQDYLKKNPNGYCHIDVNQAAYPVIDANRYTKPSDEEIKSK  240
SpMsrAB2  QPLAVEKENLKNFVVAEDYHQDYLKKNPNGYCHINVNQAAYPVIDASKYPKPSDEELKKT  237
           : :*** * *:::* **:**********.*****:**:*  *::* .:* **:* *:*.

SgMsrA    LTQEQYQVTQLSATERPFHNAYNATFEEGIYVDVTTGEPLFFAGDKFESGCGWPSFSRPI  238
SgMsrAB   LSPEEYAVTQKNDTERAFSNRYWDKFDAGIYVDVVTGEPLFSSKDKFDSGCGWPSFTRPI  300
SpMsrAB2  LSPEEYAVTQENQTERAFSNRYWDKFESGIYVDIATGEPLFSSKDKFESGCGWPSFTQPI  297
          *: *:* ***  . *** * * *  .*: *****:.****** : ***:*********:**

SgMsrA    AREVLRYYEDKSHGMERIEVRSRSGNAHLGHVFTDGPESAGGLRYCINSAALRFIPKEKM  298
SgMsrAB   SPDVATYKEDKSFNMTRTEVSRVGNSHLGHVFTDGPKDKGGLRYCINSLSIKFIPKAEM   360
SpMsrAB2  SPDVVTYKEDKSYNMTRMEVRSRVGDSHLGHVFTDGPQDKGGLRYCINSLSIRFIPKDQM  357
          : :*  * ****..* * ***** *::***********:. ********* :::**** :*

SgMsrA    EAEGYAYLLQHMK      311
SgMsrAB   EEKGYGYLLDYV-      372
SpMsrAB2  EEKGYAYLLDYVD      370
          * :**.***::: 
```

b.

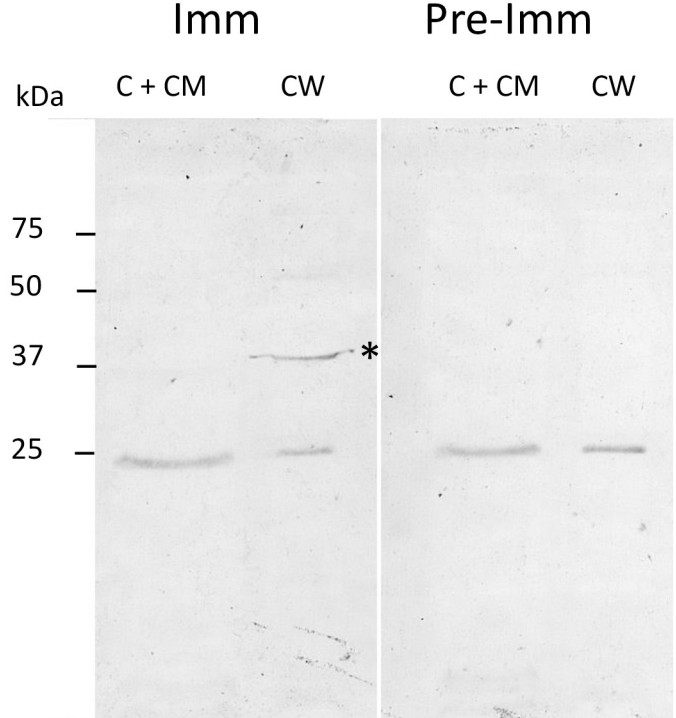

**Fig 4. Sequence alignment of MsrAB, MsrA, and MsrAB2 and cellular localization of MsrAB.** (a) Sequence alignment of *S. gordonii* MsrAB, *S. gordonii* MsrA, and *S. pneumoniae* MsrAB2 using Cluster Omega. Signal sequence is underlined. Conserved cysteine residues are boxed. The A domain of MsrAB is boxed and shaded and the B domain is boxed. (b) Cell wall (CW) and cytoplasm-cytoplasmic membrane (C +CM) fractions of *S. gordonii* were probed with anti-MsrAB antibody (Imm, 1/500) or pre-immune antibody (Pre-imm, 1/500). The 25 kDa band is likely a cross-

reaction to the goat anti-mouse secondary antibody. The intensity of this 25 kDa band is roughly equal in samples suggesting equal loading of samples.

conditions of no exogenous $H_2O_2$. The *msrAB* mutant showed no difference in total growth in biofilm cultures (Fig 5B), and a similar growth curve (Fig 5C) compared to the parent and *msrA* mutant suggesting that the lower level of biofilm formation was not because of differences in growth rates.

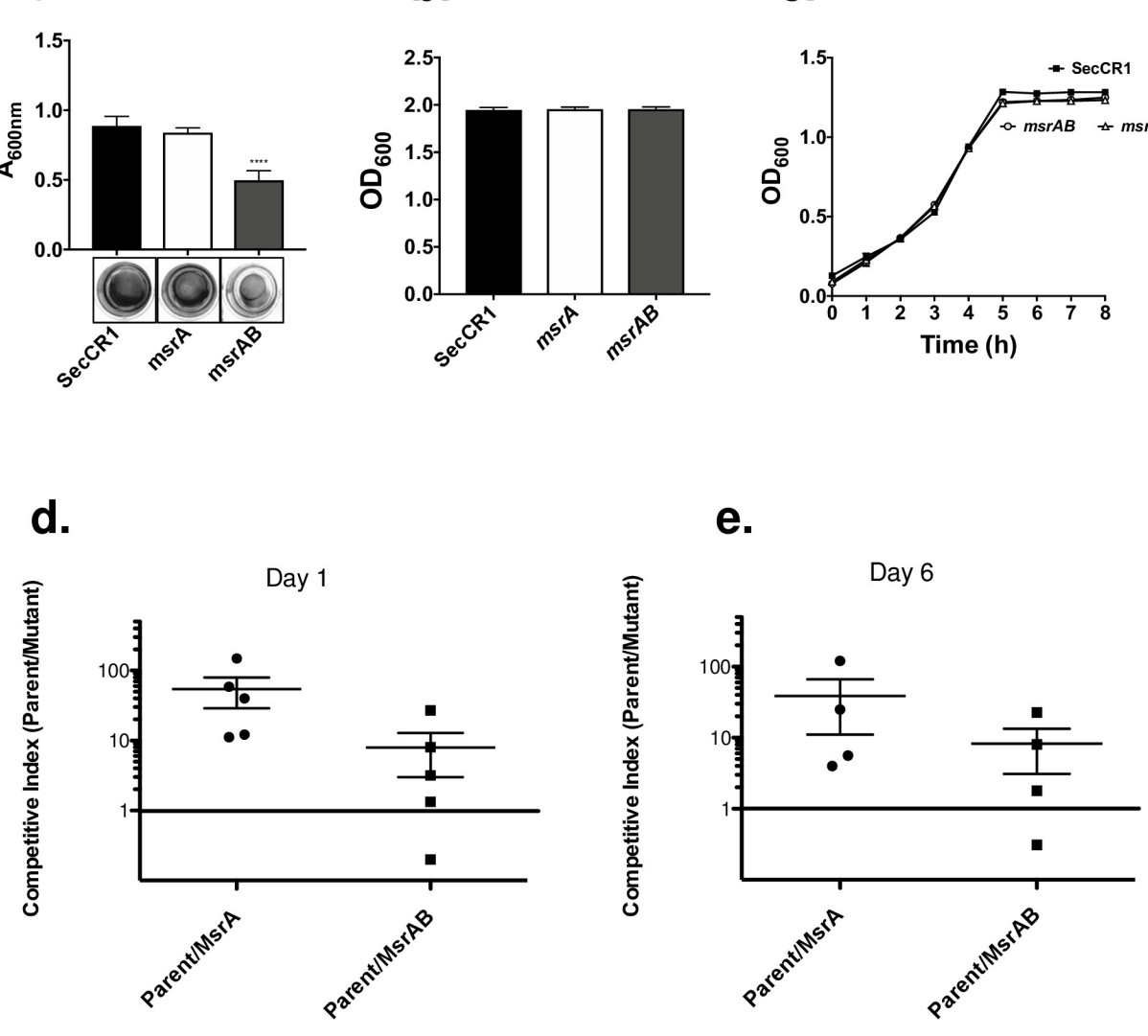

**Fig 5. Biofilm formation and oral colonization by *S. gordonii*.** (a) Biofilm formed by *S. gordonii* parent, *msrAB*, and *msrA*. Crystal violet staining of 24 h biofilms grown in 24-well plates. Results are means ± SD of three independent experiments with quadruplicate in each experiment. The lower panel shows representative wells after staining. (b) Total amount of growth of biofilm and planktonic cells. (c) Growth curves of *S. gordonii* parent, *msrAB*, and *msrA*. Results of b and c are means ± SD of three independent experiments with duplicate in each experiment. (d and e) Oral colonization by *S. gordonii* in BALB/c mice. Mice were inoculated with roughly equal amount of the parent and mutant strains. Bacteria were recovered by swabbing oral surfaces after 1 and 6 days and enumerated using selective agar. Results are reported as competitive index from each mouse. A competitive index value of one represents equal number of parent and mutant.

As an inhabitant of oral biofilms, biofilm formation is integral to *S. gordonii* colonization. In addition, it has been shown that *in vitro* biofilm formation correlates to *in vivo* colonization in mice for *S. gordonii* [16] and *S. pneumoniae* [17]. Given the findings in biofilm formation, we performed a competitive assay between the parent and *msrAB* mutant in a mouse model of oral colonization [16, 18]. We also included the *msrA* mutant for comparison. Mice were inoculated with equal amounts of the parent and mutant strains. Following colonization, bacteria were recovered from the oral cavity and enumerated by growing with antibiotics that allowed differentiation of the two strains. The competitive index was then calculated as the ratio of the parent to the mutant, where a value of one indicates no difference. Consistent with the ability of the *msrAB* mutant to form less biofilm *in vitro*, the mean competitive index for the *msrAB* mutant at days 1 and 6 were 8.0 ($P = 0.02$) and 8.2 ($P = 0.01$), respectively, representing a significant decrease compared to the parent (Fig 5D and 5E). Interestingly, the *msrA* mutant also displayed a reduced colonization ability. The mean competitive index at days 1 and 6 were 54.3 ($P = 0.001$) and 38.8 ($P = 0.003$), respectively, suggesting that the *msrA* mutant was even less fit in the competitive assay than the *msrAB* mutant (Fig 5D and 5E).

## MsrAB, SdbB, and Sgo_1177 production is induced in response to aeration and in planktonic cells

Since MsrAB, SdbB, Sgo_1177, CcdA1, and CcdA2 are involved in tolerance to $H_2O_2$ and *S. gordonii* is exposed to $H_2O_2$ during aerobic growth [2], we hypothesize that the production of these proteins is elevated to deal with aeration. To test this, *S. gordonii* was grown under aerobic and anaerobic conditions, and the level of these proteins was examined by immunoblotting. Due to a lack of antibody to CcdA1 and CcdA2, western blotting of these two proteins were not performed. The results showed that the level of MsrAB was two-fold higher when *S. gordonii* was grown aerobically compared to anaerobically. The same was observed for SdbB and Sgo_1177 (Fig 6A and 6C).

Because biofilm and planktonic cells are exposed to different $O_2$ levels, the expression of *msrAB*, *sdbB*, and *sgo_1177* may be altered in biofilms. To test this, the level of MsrAB, SdbB, and Sgo_1177 in planktonic and biofilm cells was examined. The results showed that planktonic cells, which exposed to a higher level of oxygen, produced two-fold higher MsrAB, SdbB, and Sgo_1177 than the biofilm cells (Fig 6D and 6F).

## Discussion

In this study, we reported a reducing pathway in *S. gordonii* that plays a role in oxidative stress tolerance, biofilm formation, and fitness in oral colonization in mice. The pathway consists of MsrAB as the enzyme that reduces methionine sulfoxide and two thioredoxin-like proteins and two CcdA proteins that play roles in maintaining the catalytic cycle of MsrAB.

Our results showed that inactivating the *msrAB* gene and genes within the *msrAB* operon (*sgo_1177* and *ccdA1*) and genes in the adjacent operon (*sdbB* and *ccdA2*) render *S. gordonii* sensitive to $H_2O_2$ and methionine sulfoxide suggesting that the products of these genes are involved in oxidative stress tolerance. It has been reported in other bacterial species that *msr* mutants are sensitive to $H_2O_2$ [10, 19, 20]; thus, the sensitivity of our *msrAB* mutant is consistent with those in the literature. However, the sensitivity displayed by other mutants is a bit unexpected. If these other proteins are part of a pathway involving MsrAB, then it makes sense that these mutants are also sensitive to these agents. SdbB and Sgo_1177 are thioredoxin-like proteins, and CcdA1 and CcdA2 are annotated as cytochrome *c* biogenesis proteins, and they have the ability to shuffle electrons between proteins [13, 21]. Thus, they can play a role in the

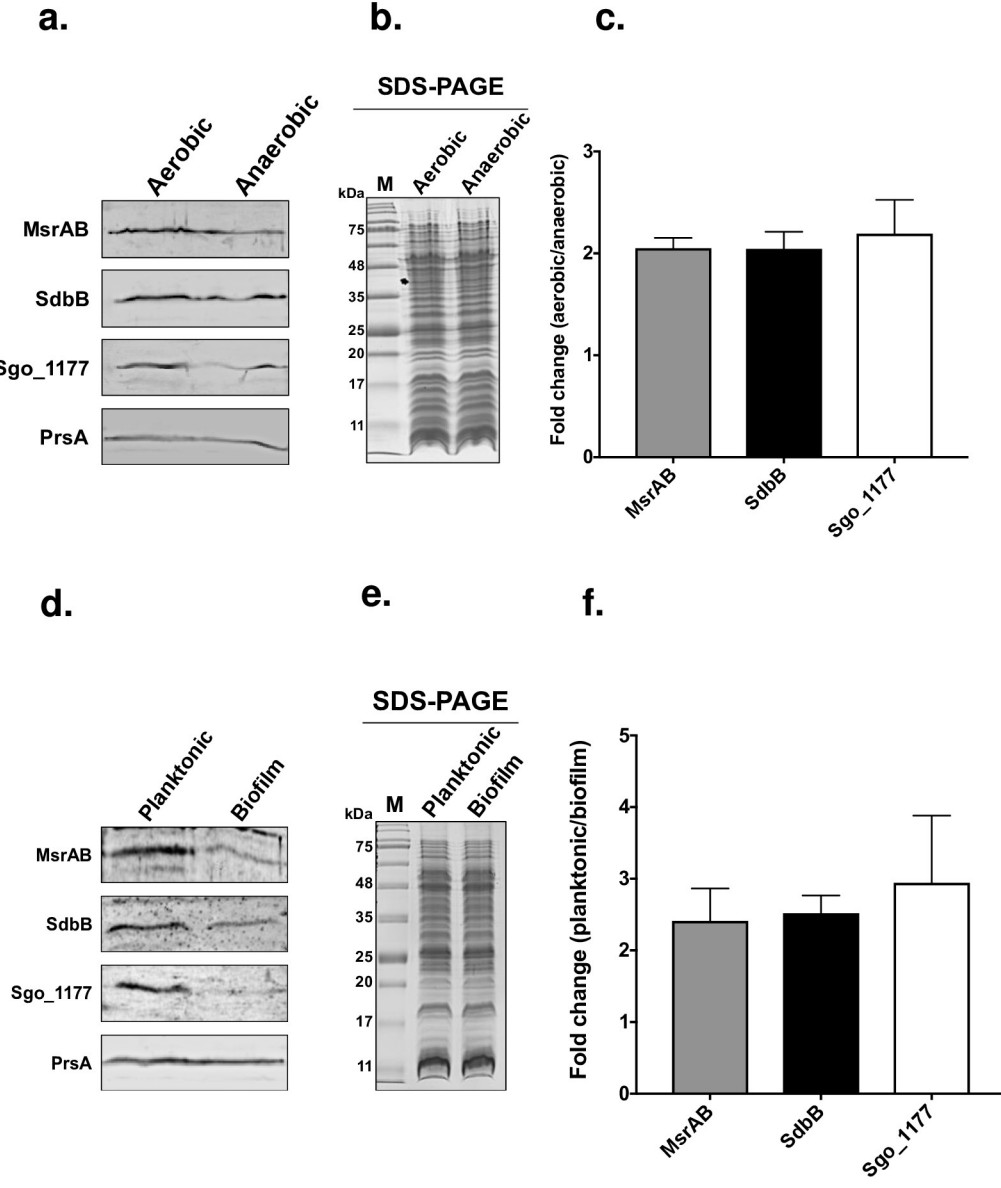

**Fig 6. Production of MsrAB, SdbB, and SdbC by *S. gordonii*.** (a) Levels of MsrAB, SdbB, and Sgo_1177 detected by specific antibodies in *S. gordonii* cultures grown under aerobic and anaerobic conditions. (b) SDS-PAGE gels showing samples used in panel a contained roughly equal amount of proteins. (c) Densitometry analysis of the MsrAB, SdbB, Sgo_1177 bands by Image J. Results are reported as the ratio of band intensity from aerobic over anaerobic conditions. (d) Levels of MsrAB, SdbB, and Sgo_1177 in *S. gordonii* planktonic and biofilm cells. (e) SDS-PAGE to show equal loading. (f) Densitometry analysis of MsrAB, SdbB, Sgo_1177 bands by Image J. Results are reported as the ratio of band intensity between planktonic and biofilm cells. Results are means ± SD of two independent experiments. In panels a and d, samples were also probed for an unrelated 35-kDa lipoprotein PrsA (anti-PrsA 1/1000) as loading controls.

reduction of methionine sulfoxide by passing electrons to MsrAB. Interruption of the electron flow should result in a methionine sulfoxide sensitivity phenotype.

The above notion is supported by the results of the methionine sulfoxide reductase assay and disulfide exchange reactions. The former clearly showed that MsrAB required SdbB or Sgo_1177 for activity, and the later showed that oxidized MsrAB was readily reduced by SdbB and Sgo_1177. These results indicate that SdbB and Sgo_1177 are immediate partners of

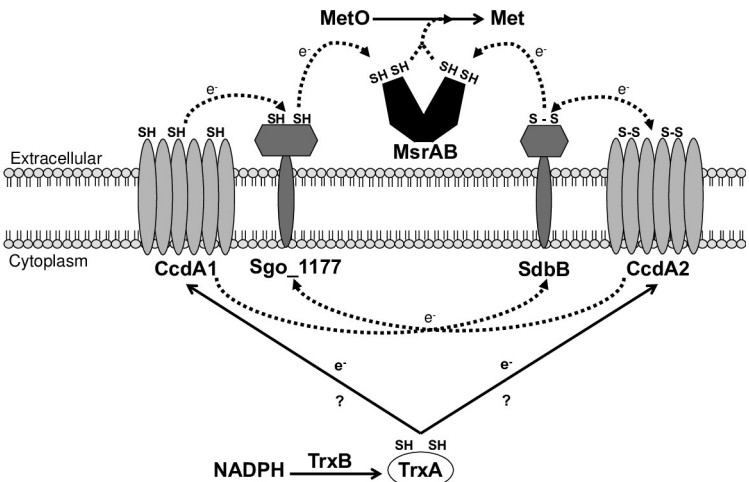

**Fig 7. A proposed reducing pathway in *S. gordonii*.** MsrAB reduces methionine sulfoxide (MetO) to methionine (Met) to repair oxidatively damaged proteins. In doing so, MsrAB is oxidized and requires SdbB or Sgo_1177 for regeneration to its active reduced form. SdbB or Sgo_1177 relies on CcdA proteins to regenerate. CcdA1 and CcdA2 are integral membrane proteins and obtain electrons from cytoplasmic NADPH via thioredoxin (TrxA) through the action of thioredoxin reductase (TrxB). The last part of the pathway has not been experimentally shown in this report and is depicted as question marks.

MsrAB involved in the regeneration of reduced MsrAB. Interestingly, the results also showed that MsrAB could not reduce methionine sulfoxide in the presence of CcdA1 or CcdA2 and oxidized MsrAB was not reduced by CcdA1 or CcdA2. Instead, SdbB and Sgo_1177 were readily reduced by CcdA1 and CcdA2 suggesting that CcdA1 and CcdA2 are immediate partners of SdbB and Sgo_1177. Collectively, these results suggest a reducing pathway consists of MsrAB as the enzyme and its catalytic cycle is maintained by SdbB and Sgo_1177 as immediate partners and CcdA1 and CcdA2 are upstream partners of SdbB and Sgo_1177 (Fig 7). The electrons needed for the reactions are likely originated from NADPH in the cytoplasm. CcdA1 and CcdA2 are predicted to be integral membrane proteins with six transmembrane regions [13]. A recent study showed that the archaeal CcdA is capable of relaying electrons from thioredoxins across the cytoplasmic membrane in a substrate-specific pathway [21] supporting the notion that CcdA1 and CCdA2 may relay electrons from the cytoplasm to SdbB and sgo_1177.

The MsrAB reducing pathway described here is similar to the one recently reported in *S. pneumoniae* [15]. In *S. pneumoniae*, the pathway is needed for oxidative stress resistance and virulence. The pathway consists of similar components, i.e. MsrAB2, two thioredoxin-like lipoproteins (Etrx1 and Etrx2) and two CcdA proteins. The genes coding for these components are in two genetic loci organized in a similar fashion to those in *S. gordonii* (S1 Fig). *In silico* analysis showed that homologs of SdbB, Sgo_1177 and CcdA proteins are also present in *Streptococcus sanguinis* and *Clostridium botulinum* suggesting that the reducing pathway is not limited to *S. gordonii* and *S. pneumoniae* (S1 Fig). We noted one difference between the pathway in *S. gordonii* and *S. pneumoniae* is the specificity of the two-thioredoxin-like proteins. In *S. gordonii*, there is virtually no difference between SdbB and Sgo_1177 for the MsrAB "A" and "B" domain, while in *S. pneumoniae*, Etrx1 prefers the A domain while Etrx2 has similar specificity for both the A and B domains [15].

Our immunoblotting results showed that MsrAB is an extracytoplasmic enzyme. MsrAB2 in *S. pneumoniae* is also reported to be an extracytoplasmic enzyme [15]. Sequence analysis showed that MsrAB and MsrAB2 contain a signal peptide. This is in contrast to the previously described MsrA of *S. gordonii*, which does not carry a signal peptide and was found mostly in

the cytoplasm [10]. The difference in cellular location suggests that the role of MsrA is to protect cytoplasmic proteins while that of MsrAB is to protect extracytoplasmic proteins, such as surface adhesins important to biofilm formation. Perhaps, the biofilm formation results reflect that. Nonetheless, both mutants shared the feature of conferring resistance to $H_2O_2$ and methionine sulfoxide. It is not uncommon for a bacterial species to have multiple methionine sulfoxide reductases. *S. pneumoniae* has a cytoplasmic MsrAB1 in addition to the extracytoplasmic MsrAB2 [15, 22]. *Escherichia coli* possesses four cytoplasmic and one periplasmic methionine sulfoxide reductases [23–27]. These methionine sulfoxide reductases not only reside in different cellular compartments but also exhibit different substrate specificity. For example, MsrAB1 has a preference for the Met-*S*-O isomer over the Met-*R*-O isomer of methionine sulfoxide [22]. At this time, it is unclear whether MsrA and MsrAB of *S. gordonii* have any differences in substrate specificity.

One interesting feature of the reducing pathway in *S. gordonii* is the redundancy of the components in the pathway. SdbB and Sgo_1177 share a 42% sequence identity and appear to be able to carry out a similar function, i.e. donate electrons to MsrAB. MsrAB exhibited an identical specific activity in the presence of SdbB or Sgo_1177. As well, the A and B domains of MsrAB exhibited the same specificity activity in the presence of SdbB or Sgo_1177. These results suggest that SdbB and Sgo_1177 are redundant in MsrAB activity. However, we recently reported that SdbB is a redox partner of SdbA and Sgo_1177 does not have such a function [13]. CcdA1 and CcdA2 are also highly homologous with 69.8% sequence identity; both were able to reduce SdbB and Sgo_1177. CcdA2, but not CcA1, can function as a redox partner of SdbA [13]. Thus, these proteins have overlapping functions in the MsrAB reducing pathway but not in the SdbA oxidative pathway.

The expression of MsrAB, SdbB, and Sgo_1177 was higher under aerobic growth condition and planktonic lifestyle. This is likely due to the fact that oxidative stress is higher under these growth conditions and *S. gordonii* needs to elevate the level of these proteins to counter the stress. Similar findings were reported in other bacteria, including *E. coli* and *Corynebacterium glutamicum* [19, 20, 28].

It is interesting to note that the *msrAB* mutant was defective in biofilm formation and outcompeted by the parent in a mouse oral colonization assay. In a previous study, the disulfide bond oxidase *sdbA* mutant of *S. gordonii* formed more biofilm *in vitro* and also outcompeted the parent *in vivo* [16]. In the case of the *msrA* mutant, the *in vitro* biofilm formation ability was not altered, but it was outcompeted by the parent in oral colonization. These findings underscored the point that *in vitro* biofilm results do not always predict the outcome of *in vivo* assay.

In conclusion, MsrAB, SdbB, Sgo_1177, CcdA1, and CcdA2 form a reducing pathway in *S. gordonii* that is needed for oxidative stress tolerance, biofilm formation, and oral colonization in mice. This is the first report of a reducing pathway in *S. gordonii*.

## Materials and methods

### Bacterial strains and growth conditions

Bacterial strains used in this study are listed in Table 1. *S. gordonii* SecCR1 was used as the parent strain [29]. *S. gordonii* strains were routinely grown at 37˚C, 5% $CO_2$ in HTVG (per ml: 5 mg of glucose, 35 mg of tryptone, 0.04 μg of *p*-aminobenzoic acid, 0.2 μg of thiamine- HCl, 1 μg of nicotinamide, and 0.2 μg of riboflavin, 100 mM HEPES, pH 7.6) [30], or in tryptic soy broth, TYG (per 100 ml: 1 g of tryptone, 0.5 g of yeast extract, 0.2 g of glucose, 0.3 g of $K_2HPO_4$), or biofilm medium [31] as indicated. *Escherichia coli* was grown in Luria-Bertani

(LB) broth (per 100 ml: 1 g tryptone, 1 g NaCl, 0.5 g yeast extract) with shaking (180 rpm) at 37˚C.

When needed, antibiotics were used at the following concentrations: for *S. gordonii*, spectinomycin (250 μg/ml); kanamycin (250 μg/ml); tetracycline (10 μg/ml); erythromycin (10 μg/ml); chloramphenicol (5 μg/ml); and for *E. coli*, ampicillin (100 μg/ml); kanamycin (50 μg/ml); and tetracycline (10 μg/ml).

## Genetic manipulation

Gene knockout mutants were constructed by insertional inactivation using the erythromycin resistance cassette (*ermAM*) [32], or the kanamycin resistance cassette (*aphA3*) [33] as described previously [13]. Briefly, the antibiotic resistance cassette and the upstream and downstream regions of the target gene were amplified by polymerase chain reaction (PCR) using primers listed in S1 Table, digested with the appropriate restriction enzymes, and ligated using T4 DNA ligase. The ligation products were amplified using the outside primers, and the resulting PCR products were transformed into *S. gordonii* SecCR1. The insertion of the antibiotic resistance cassette into the target gene was confirmed by PCR and gene inactivation was confirmed by immunoblotting (S4 Fig).

The *sdbBccdA2*-complemented mutant was constructed previously [13]. In this mutant, a functional copy of *sdbB* and *ccdA2* genes was knockback into the chromosome of the *sdbBccdA2* mutant.

The *sgo_1177ccdA1*-complemented mutant was constructed as follows. First, the downstream fragment of *ccdA1sgo_1177msrAB* operon carrying a portion of *msrAB* and *sgo_1175* was amplified using primers SL1147/SL1148 and digested with BamHI. The *cat* resistance cassette was cut from pCopCAT/pUC18 using SphI and BamHI [34]. The downstream fragment was then ligated to the *cat* resistance cassette, and the ligation product was amplified using phusion DNA polymerase and the outside primers SL666/SL1148. The PCR product of the *cat*-downstream fragment was then digested with HindIII. Next, the *msrAB* entire gene was amplified using primers SL1319/SL1146 and digested with HindIII. The *cat*-downstream and the *msrAB* genes were ligated together using T4 DNA ligase. The ligation product was amplified using phusion DNA polymerase and the outside primers SL1146/SL1148 and digested with XbaI, a unique restriction site within the *msrAB* gene. Next, the *sgo_1180ccdA1sgo_1177msrAB* operon was amplified using primers SL1050/SL1319 and digested with XbaI. Following gel purification, the *sgo_1180ccdA1sgo_1177msrAB* operon was ligated with *msrAB-cat*-downstream fragment using T4 DNA ligase, and the ligation product was amplified using SL1050/SL1148. The PCR product was transformed into the *sgo_1177ccdA1* mutant, and the transformants were selected on brain heart infusion agar containing chloramphenicol and replica-plated to identify kanamycin- and erythromycin-sensitive colonies and confirmed by PCR.

## $H_2O_2$ killing assay

The $H_2O_2$ killing assay was performed as described previously with modifications [15]. Briefly, overnight cultures of *S. gordonii* were diluted 1:40 (v/v) into 4 ml of pre-warmed freshly made HTVG and grown to an $OD_{600}$ of 0.2. An aliquot of the cultures was treated with 10 mM $H_2O_2$ at 37˚C, 5% $CO_2$. In parallel, another aliquot of the culture was not treated and served as the control. After 30 min, the cultures were serially diluted and plated in triplicate onto TYG agar plates. The plates were incubated for 24 h and colonies were counted. The percentage of survival was calculated as (CFU of $H_2O_2$-treated sample / CFU of the untreated sample) x 100.

## Methionine sulfoxide sensitivity assay

Overnight cultures were diluted 1:20 into 5 ml of pre-warmed tryptic soy broth and grown to $OD_{600} = 0.1$. Aliquots (4 μl) of the culture were used to inoculate 200 μl of pre-warmed tryptic soy broth with and without 20 mM methionine sulfoxide in 96-well flat-bottom microplates (Corning Costar, Fisher Scientific, Ontario, Canada). The plates were incubated for 24 h and $OD_{600}$ was measured using a microplate reader.

## Cell fractionation

*S. gordonii* SecCR1 was grown in 100 ml HTVG to late exponential phase of growth. Cells were harvested by centrifugation (10 000 x g, 10 min) and resuspended in 1 ml of phosphate-buffered saline (PBS, 8.7 mM $Na_2HPO_4$, 1.5 mM $NaH_2PO_4$, 1.45 M NaCl, pH 7.2) in a 2 ml screw-cap tube (Sarstedt). Glass beads (0.25 ml, 400 μm, VWR International) were added to the cell suspension, and the cells were broken by homogenization (7 bursts of 45 sec each with 1 min cooling between each burst, setting #4,) using a Fast-Prep homogenizer (Thermo Savant, Model FP120). Following homogenization, the suspension was centrifuged (10 s, 10, 000 x g) to sediment the glass beads, and the supernatant was saved. PBS (0.25 ml) was added to the glass beads and pulse-centrifuged as above. The supernatant was pooled with the first supernatant. The supernatant was centrifuged (2,000 x g, 3 min) to remove unbroken cells. One ml of the supernatant was centrifuged at 20,000 x g for 20 min. The supernatant was saved as the cytoplasm-cell membrane fraction. The pellet was resuspended in 0.2 ml PBS as the cell wall fraction. Twenty μl of cytoplasm-cell membrane fraction and 4 μl of cell wall, representing the proportional quantity of cell suspension, were analyzed by western blotting.

## Expression of MsrAB, SdbB, and Sgo_1177 in cultures

An overnight culture (1 ml) of *S. gordonii* SecCR1 grown stationary in HTVG under anaerobic conditions (GasPak, anaerobic gas generating system, BD) was added to 9 ml of pre-warmed HTVG, and the culture was incubated at 37˚C under anaerobic conditions. A second overnight culture (1 ml) grown stationary in HTVG in a $CO_2$ incubator was added to 9 ml of pre-warmed HTVG, and the culture was incubated aerobically in ambient atmosphere at 37˚C on a shaker (20 rpm). When the cultures reached $OD_{600} = 0.8$, cells were harvested by centrifugation and analyzed by western blotting for MsrAB, SdbB, and Sgo_1177.

To test the level of expression of MsrAB, SdbB, and SdbC in biofilm and planktonic cells, overnight cultures of *S. gordonii* SecCR1 grown in HTVG were centrifugation (3000 x g for 10 min), and the cells were suspended in biofilm medium to an $OD_{600} = 0.25$. The cell suspension (1 ml/well) was used to inoculate a 24-well flat-bottom plate (Costar). The plates were incubated at 37˚C, 5% $CO_2$. Next day, the supernatant containing the planktonic cells was collected, and the wells were washed once with 1 ml of sterilized PBS to remove loosely attached cells. The biofilm cells were resuspended in biofilm medium (1ml/well) by vigorous pipetting. The planktonic and biofilm cell suspension were adjusted to $OD_{600} = 0.8$ and cells were then harvested by centrifugation and analyzed by western blotting.

## Recombinant proteins

Recombinant proteins (SdbB, Sgo_1177, MsrAB, A domain of MsrAB, B domain of MsrAB, and TrxB) were expressed as in-frame N-terminal $His_6$ fusion proteins using the vector pQE30 (Qiagen). Briefly, DNA coding for the mature protein was PCR amplified using primers listed in S1 Table. The PCR products and pQE30 were digested and ligated with T4 DNA ligase. The ligation products were transformed to *E. coli*. SdbB, Sgo_1177, TrxB, MsrAB, and A domain of

 

MsrAB were purified from the soluble fractions of cell lysates using His$_{60}$ Ni Superflow columns (Clontech) according to manufacturer's instructions. The B domain of MsrAB was found to be insoluble. To purify this protein, the insoluble fraction of cell lysates was solubilized with 8 M urea and purified using an on-column renaturing protocol described previously [35]. Briefly, urea-solubilized samples were applied to a His$_{60}$ Ni Superflow column. After washing to remove unwanted proteins, the column was sequentially washed with refolding buffers (300 mM NaCl, 10 mM imidazole,1 mM oxidized glutathione, 3 mM reduced glutathione, 50 mM NaH$_2$PO$_4$H$_2$O, pH 7) containing decreasing concentrations (6 M, 4 M, 2 M, 1 M, and 0 M) of urea. The B domain was eluted with 300 mM imidazole.

The cloning and purification of recombinant CcdA2 were described previously [13]. CcdA1 was cloned and purified exactly as described for CcdA2.

Anti-MsrAB antibodies were raised in BALB/c mice using methods similar to those described previously for anti-SdbB antibodies [13].

To prepare fully oxidized proteins, samples were incubated in the oxidizing buffer (50 mM oxidized glutathione, 100 mM Tris-HCl, pH 8.8, 200 mM KCl, 1 mM EDTA) for 1 hour at room temperature. Samples were then dialyzed against 100 mM sodium phosphate buffer (pH 7) to remove the unreacted glutathione. CcdA1 and CcdA2 were oxidized similarly but in the presence of 3 mM *n*-dodecylphosphocholine (DPC) (Anatrace). Excess glutathione was removed from CcdA1 and CcdA1 using desalting columns PD-10 (Sephadex G-25 column; GE Life Sciences) equilibrated with 50 mM sodium phosphate buffer (pH 7) containing 3 mM DPC.

To fully reduce the purified proteins, samples were incubated with 50 mM DTT for 30 minutes on ice. After incubation, DTT was removed by passing the samples through PD-10 columns [36]. Reduced CcdA1 and CcdA2 were similarly prepared but in the presence of 3 mM DPC. The complete oxidation and reduction of the recombinant proteins were confirmed by reaction with Ellman's reagent 5,5'-dithio-bis(2-nitrobenzoic acid) (DTNB) (Sigma) and by alkylation with maleimide-PEG$_2$-biotin (Sigma) [13, 37].

## Disulfide exchange reactions

The disulfide exchange reactions between MsrAB and SdbB or Sgo_1177 were performed as described previously with modifications [13, 35]. Briefly, an equimolar of the oxidized and reduced proteins (10 μM) was incubated in 1 ml of 50 mM sodium phosphate buffer (pH 8.0) with 100 mM NaCl, and 0.5 mM EDTA at room temperature. Aliquots (200 μl) of the reaction were transferred to new tubes at specified times, and the reaction was terminated by TCA precipitation. Samples were then alkylated with 5 mM maleimide-PEG$_2$-biotin (525 Da), and excess maleimide-PEG$_2$-biotin was removed by TCA precipitation and acetone wash. Pellets were then resuspended in 100 mM Tris- HCl (pH 7.0) containing 1% (w/v) SDS and analyzed by Western blotting.

Disulfide exchange reactions between CcdA proteins and MsrAB, SdbB or Sgo_1177 were performed as described above but in the presence of 4.5 mM DPC and 1 μM of each protein. The mouse anti-MsrAB (1/500), anti-SdbB (1/1000) [13], and anti-Sgo_1177 (1/1000) [13] antisera were used for Western blot analysis.

## Methionine sulfoxide reductase assay

The methionine sulfoxide reductase activity was evaluated by monitoring the oxidation of NADPH [27]. The assays were performed at room temperature in 100 μl volumes in 96-well plates (model 3635, UV transparent flat bottom microplate, CORNING). Briefly, 10 μM of reduced MsrAB was added to a solution containing 4 μM reduced thioredoxin reductase TrxB,

250 μM NADPH (Sigma Aldrich), 100 mM methionine sulfoxide (Sigma Aldrich), 1 mM EDTA, 50 mM Tris-HCl (pH 7.5) and 40 μM of the test protein (SdbB, Sgo_11777, CcdA1, or CcdA2). The oxidation of NADPH was monitored at $A_{340\text{ nm}}$ using a microplate reader (Synergy HT; BioTeK, USA). Samples with no test protein (SdbB, Sgo_1177, CcdA1, or CcdA2), no MsrAB, or no methionine sulfoxide were used as negative controls.

## Biofilm formation

Biofilms were grown as described previously [14]. Briefly, cells from *S. gordonii* late exponential cultures in HTVG ($OD_{600}$ = 1.2) were suspended in biofilm medium to an $OD_{600}$ of 0.250 and seeded 1 ml per well onto flat-bottom 24-well plates (Falcon, model #353047). The plates were incubated for 24 h at 37°C, 5% $CO_2$. Following incubation, the medium was removed, and the wells were washed twice with 1 ml phosphate-buffered saline (PBS) to remove loosely attached cells. The plates were air-dried for 15 min and fixed with 10% (v/v) formaldehyde and 5% (v/v) acetic acid in PBS for 15 min and were washed 3 times with 1 ml PBS. The biofilms were then stained with 0.5 ml of 0.1% crystal violet for 15 min. The wells were rinsed three times, and the bound stain was solubilized in 1 ml of acetone/ethanol solution (1:1). The liquid was transferred to a new microtiter plate, and $A_{600}$ was measured in a microplate reader. The biofilm assays were carried out in triplicate with three or more separate experiments.

Total growth of biofilm and planktonic cells were prepared by vigorous pipetting to remove attached cells from bottom of the well. The optical density of the combined biofilm and planktonic cells was measured at 600 nm for triplicate wells for each strain. Crystal violet staining confirmed that the biofilms were successfully detached.

## Competitive oral colonization in mice

The animal work in this study was approved by the ethics committee (University Committee on Laboratory Animals of Dalhousie University). Animals were given plenty of foods and water as well as normal environmental enrichment devices (toys) to alleviate sufferings. Oral colonization was tested as described previously [14, 18]. BALB/c mice (female, 4 weeks old, $n$ = 9 per group) were fed kanamycin (500 μg/ml) in drinking water for 2 days prior to colonization. The *S. gordonii* parent and mutant strains were grown for 18 h and then dechained by vigorously passing the cells through a 26 G needle. Microscopy was used to confirm dechaining. Cultures were standardized by their optical density, and the parent and the mutant were then mixed in a 1:1 ratio (50 μl total volume). The inocula were plated for enumeration, and the parent/*msrA* mix contained 0.9 x $10^9$ CFU/ml of the parent and 7.3 x $10^9$ CFU/ml of the *msrA* mutant. The parent/*msrAB* mix contained 2.5 x $10^9$ CFU/ml of the parent and 3.3 x $10^9$ CFU/ml of the *msrAB* mutant.

The mice were sedated with ketamine-xylazine and inoculated with 10 μl intranasally and 40 μl orally with the bacteria mixture. After 1 and 6 days, cohort of mice were euthanized by isoflurane inhalation and subsequent $CO_2$, and sterile swabs were used to recover bacteria from the oral cavity. Swabs were mixed with brain heart infusion broth by vortexing and then aliquots plated. The bacteria were plated on brain heart infusion agar containing spectinomycin or erythromycin that allowed for differentiation of the parent and mutant. The competitive index was calculated as the ratio of the CFU/ml of the parent to mutant recovered at days 1 and 6, divided by the ratio in the inoculum.

## Statistical analysis

The results were analyzed using one-way ANOVA between the parent and mutant. A *P* value of < 0.05 was considered statistically significant.

## Supporting information

**S1 Fig. Genetic organization of *sgo_1177ccdA1msrAB* and *sdbBccdA2* operons in *S. gordonii* and other Gram-positive bacteria.**
(PDF)

**S2 Fig. SDS-PAGE of recombinant MsrAB, SdbB, Sgo_1177, CcdA1, CcdA2, TrxB, A domain of MsrAB, and B domain of MsrAB.** M: Prestained protein markers.
(PDF)

**S3 Fig. Lack reduction of methionine sulfoxide when MsrAB or methionine sulfoxide (MetO) are omitted in reactions.**
(PDF)

**S4 Fig. Immunoblotting of *S. gordonii* mutant strains confirming the gene-knockout phenotypes.** Antibodies used were mouse sera: anti-SdbB (1/500), anti-Sgo_1177 (1/1000), anti-MsrAB (1/500), and anti-PsrA (1/2000). PrsA is an unrelated lipoprotein and served as a loading control. CBB: Coomassie-blue stained SDS-PAGE gels showing amount of proteins in samples.
(PDF)

**S5 Fig. Original blots and gels.**
(PDF)

**S1 Table. Primers used in this study.**
(PDF)

## Author Contributions

**Conceptualization:** Song F. Lee.

**Data curation:** Naif Jalal.

**Formal analysis:** Naif Jalal, Song F. Lee.

**Funding acquisition:** Song F. Lee.

**Investigation:** Naif Jalal.

**Supervision:** Song F. Lee.

**Writing – original draft:** Naif Jalal.

**Writing – review & editing:** Naif Jalal, Song F. Lee.

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
