## [Decision Letter · Decision Letter 0]

17 Jan 2020

PONE-D-19-35806

The MsrAB Reducing Pathway of Streptococcus gordoniiis Needed for Oxidative Stress Resistance, Biofilm Formation, and Oral Colonization in Mice

PLOS ONE

Dear Dr. Lee,

Thank you for submitting your manuscript to PLOS ONE. It has been reviewed by two experts in the area. Both reviewers thought the manuscript addresses an interesting topic in oxidative stress tolerance response and biofilm formation by a major oral streptococcus, and the findings are novel. After careful consideration, we feel the manuscript is well written, but as detailed in the reviewers' comments, there are some minor issues that will require some minor modifications in order to improve the readability. Therefore, we invite you to submit a revised version of the manuscript that addresses the points raised during the review process.

We would appreciate receiving your revised manuscript by 20 days. To enhance the reproducibility of your results, we recommend that if applicable you deposit your laboratory protocols in protocols.io, where a protocol can be assigned its own identifier (DOI) such that it can be cited independently in the future. For instructions see: http://journals.plos.org/plosone/s/submission-guidelines#loc-laboratory-protocols

We look forward to receiving your revised manuscript.

Kind regards,

Z. Tom Wen, PhD, DVM

Academic Editor

PLOS ONE

Journal Requirements:

2. As part of your revision, please complete and submit a copy of the ARRIVE Guidelines checklist, a document that aims to improve experimental reporting and reproducibility of animal studies for purposes of post-publication data analysis and reproducibility: https://www.nc3rs.org.uk/arrive-guidelines. Please include your completed checklist as a Supporting Information file. Note that if your paper is accepted for publication, this checklist will be published as part of your article.

3. To comply with PLOS ONE submissions requirements, in your Methods section, please provide additional information on the animal research and ensure you have included details on (1) methods of sacrifice, (2) methods of anesthesia and/or analgesia, and (3) efforts to alleviate suffering.

4. PLOS requires an ORCID iD for the corresponding author in Editorial Manager on papers submitted after December 6th, 2016. Please ensure that you have an ORCID iD and that it is validated in Editorial Manager. To do this, go to ‘Update my Information’ (in the upper left-hand corner of the main menu), and click on the Fetch/Validate link next to the ORCID field. This will take you to the ORCID site and allow you to create a new iD or authenticate a pre-existing iD in Editorial Manager. Please see the following video for instructions on linking an ORCID iD to your Editorial Manager account: https://www.youtube.com/watch?v=_xcclfuvtxQ.

Additional Editor Comments (if provided):

Reviewers' comments:

Reviewer's Responses to Questions

**Comments to the Author**

1. Is the manuscript technically sound, and do the data support the conclusions?

Reviewer #1: Yes

Reviewer #2: Yes

2. Has the statistical analysis been performed appropriately and rigorously? 

Reviewer #1: I Don't Know

Reviewer #2: Yes

3. Have the authors made all data underlying the findings in their manuscript fully available?

Reviewer #1: Yes

Reviewer #2: No

4. Is the manuscript presented in an intelligible fashion and written in standard English?

Reviewer #1: Yes

Reviewer #2: Yes

5. Review Comments to the Author

Reviewer #1: The manuscript describes the MsrAB reducing pathway of Streptococcus gordonii that affects several traits of this species, especially oxidative stress response, biofilm formation in vitro, and mice oral colonization.

The experiments are elegantly designed, the text is well-written, and the figures illustrate the findings adequately. However, there are a couple of minor points that should be clarified, as follow:

- Please clarify why the terminology “oxidative stress resistance” should be used instead of “oxidative stress tolerance”.

- Line 197. Please revise the Figure citation. It appears that the correct citation would be Fig 5d and 5e, not Fig 5b.

- Lines 197-199. Please consider adding the graphs for msrA mutant in the figure.

- Lines 232-234. The following sentence is a bit awkward and should be revised: "If one reasons that these other proteins are part of a pathway involving MsrAB, then it makes sense that these mutants are also sensitive to these agents."

- Line 390: could add the pH and the composition of PBS (e.g., with or without sodium).

- Line 513: Please include the number of mice tested per experimental group for the competitive oral colonization in mice.

- Lines 531-533. Please revise the statistical analysis description because, for example, only two groups were compared (parental versus mutant strain) for mice oral colonization assay.

Reviewer #2: Authors investigate S. gordonii MsrAB gene, and elucidated its role in oxidative stress tolerance, biofilm formation and colonization in an animal model. A new pathway is proposed involving the MsrAB gene and genes in the adjacent operons. The study is very interesting based on its novelty in the context of S. gordonii. The hypothesis are tested with rigorous experiments and the data is presented clearly.

Following are some minor concerns

In the introduction, (line 51) perhaps would be better to give a little more background information about S. gordonii oral colonization and its role in dental plaque formation.

Indicate whether MsrAB gene is an extension of the MsrA gene sequence. Is MsrAB co-transcribed? Otherwise it is a bit confusing as it says that MsrA is a cytoplasmic protein and MsrAB is an extracytoplasmic protein. MsrA also seem to be sensitive to H2O2 and affects biofilm formation which are similar properties as MsrAB. Therefore it would be better to explain a bit more clearly the differences between MsrA and MsrAB.

Fig 1a – any explanation for MsrAB to have higher percentage of survival than MsrA.

Line 127- indicates data not shown. Please check with journal guidelines regarding policy for data availability.

Has the biofilm formation been tested under H2O2 and methionine sulfoxide conditions.

Please indicate the sample size of mice used for the experiment.

6. PLOS authors have the option to publish the peer review history of their article (what does this mean?). If published, this will include your full peer review and any attached files.

Reviewer #1: No

Reviewer #2: No

---

## [Author Response · Author response to Decision Letter 0]

2 Feb 2020

Point-by-point response to reviews

Reviewer #1: The manuscript describes the MsrAB reducing pathway of Streptococcus gordonii that affects several traits of this species, especially oxidative stress response, biofilm formation in vitro, and mice oral colonization.

The experiments are elegantly designed, the text is well-written, and the figures illustrate the findings adequately. However, there are a couple of minor points that should be clarified, as follow:

- Please clarify why the terminology “oxidative stress resistance” should be used instead of “oxidative stress tolerance”.

Response: Oxidative stress tolerance is a better term. It is now used.

- Line 197. Please revise the Figure citation. It appears that the correct citation would be Fig 5d and 5e, not Fig 5b.

Response: Done. (l. 258)

- Lines 197-199. Please consider adding the graphs for msrA mutant in the figure.

Response: msrA was in the figure, i.e. first set of data in Fig 5d and 5e.

- Lines 232-234. The following sentence is a bit awkward and should be revised: "If one reasons that these other proteins are part of a pathway involving MsrAB, then it makes sense that these mutants are also sensitive to these agents."

Response: The sentence has been modified (l. 303-304).

- Line 390: could add the pH and the composition of PBS (e.g., with or without sodium).

Response: pH and composition added (l. 475).

- Line 513: Please include the number of mice tested per experimental group for the competitive oral colonization in mice.

Response: given now (l. 601).

- Lines 531-533. Please revise the statistical analysis description because, for example, only two groups were compared (parental versus mutant strain) for mice oral colonization assay.

Response: The sentence has been revised to state “between the parent and mutant” (l. 620).

Reviewer #2: Authors investigate S. gordonii MsrAB gene, and elucidated its role in oxidative stress tolerance, biofilm formation and colonization in an animal model. A new pathway is proposed involving the MsrAB gene and genes in the adjacent operons. The study is very interesting based on its novelty in the context of S. gordonii. The hypothesis are tested with rigorous experiments and the data is presented clearly.

Response: Thank you for the kind words.

Following are some minor concerns

In the introduction, (line 51) perhaps would be better to give a little more background information about S. gordonii oral colonization and its role in dental plaque formation.

Response: A couple of sentences have been added (l. 49-52).

Indicate whether MsrAB gene is an extension of the MsrA gene sequence. Is MsrAB co-transcribed? Otherwise it is a bit confusing as it says that MsrA is a cytoplasmic protein and MsrAB is an extracytoplasmic protein. MsrA also seem to be sensitive to H2O2 and affects biofilm formation which are similar properties as MsrAB. Therefore it would be better to explain a bit more clearly the differences between MsrA and MsrAB.

Response: MsrAB is a single polypeptide and the gene is located at a separate locus from msrA. This has been added to the introduction as well as the gene ID for the two genes (l. 64, l. 71-74). Hopefully, it helps to prevent confusion. 

Fig 1a – any explanation for MsrAB to have higher percentage of survival than MsrA.

Response: May be this is because MsrA, as a cytoplasmic enzyme, it plays a role in protecting essential cellular proteins; while MsrAB, as an extracytoplasmic enzyme, it protects mainly surface proteins.

Line 127- indicates data not shown. Please check with journal guidelines regarding policy for data availability.

Response: Data now included as S3 Fig in supplementary file.

Has the biofilm formation been tested under H2O2 and methionine sulfoxide conditions.

Response: No, we did not.

Please indicate the sample size of mice used for the experiment.

Response:. Number of mice per group is now given (l. 601).

---

## [Editor Report · Decision Letter 1]

6 Feb 2020

The MsrAB Reducing Pathway of Streptococcus gordoniiis Needed for Oxidative Stress Tolerance, Biofilm Formation, and Oral Colonization in Mice

PONE-D-19-35806R1

Dear Dr. Lee,

We are pleased to inform you that your manuscript has been judged scientifically suitable for publication and will be formally accepted for publication once it complies with all outstanding technical requirements. Congratulations!

With kind regards,

Z. Tom Wen, PhD, DVM

Academic Editor

PLOS ONE
---

## [Editor Report · Acceptance letter]

12 Feb 2020

PONE-D-19-35806R1 

The MsrAB Reducing Pathway of *Streptococcus gordonii* is Needed for Oxidative Stress Tolerance, Biofilm Formation, and Oral Colonization in Mice 

Dear Dr. Lee:

I am pleased to inform you that your manuscript has been deemed suitable for publication in PLOS ONE. Congratulations! Your manuscript is now with our production department. 

With kind regards,

on behalf of

Dr. Z. Tom Wen 

Academic Editor

PLOS ONE